# OpenReview forum: "Adversarial Déjà Vu: Jailbreak Dictionary Learning for Stronger Generalization to Unseen Attacks"
_ICLR.cc/2026/Conference — ICLR 2026 Poster_

### Official Review · Reviewer_5oss · 2025-10-30

**Soundness:** 3
**Presentation:** 3
**Contribution:** 3
**Rating:** 6
**Confidence:** 3

**Summary:**

The paper explores whether new jailbreak attacks primarily arise as recombinations of earlier adversarial tactics. The authors compile a large set of original–mutated prompt pairs from multiple jailbreak studies and automatically extract thousands of textual skills using a frontier LLM. These are then refined with another LLM to produce clear, human-readable skill primitives and to remove near-duplicates. They test a temporal-cutoff hypothesis, examining whether skills from newer attacks can be represented as sparse compositions of older primitives. The skill explanations are embedded using text-embedding-3-large, followed by dictionary learning and sparse coding along a Pareto knee. The resulting dictionary atoms are decoded into interpretable names through basis-pursuit attribution, LLM synthesis, and a similarity-graph redundancy filter. Evaluated on LLaMA-3.1-Instruct and Zephyr-Instruct, the proposed ASCoT framework achieves lower StrongReject harmfulness scores on previously unseen attacks.

**Strengths:**

1. The pre/post split and comparison of a compact dictionary (D_{\text{final}}) against the overcomplete Dover demonstrate high explainability with far fewer atoms and consistent sparsity across unseen families.

2. The generation protocol (composing named primitives with a helper LLM) aligns the defense with the claimed structure of jailbreaks; the dataset composition is enumerated with sizes and sources.

3. The coverage dividend and (k)-depth crossover figures isolate interesting phenomena at constant data size, supporting the skill-space framing.

**Weaknesses:**

1. The paper extracts skills, names atoms, runs redundancy decisions, and scores explainability with GPT-4.1; while human-in-the-loop verification is mentioned, the extent and protocol are not quantified, raising leakage/bias concerns. Independent human ratings (with inter-rater agreement) or a distinct non-OpenAI judge would strengthen claims.

2. Key controls are absent: (i) training on the same number of randomly composed skills vs. learned primitives; (ii) training with the same data budget but more plain harmful data; (iii) swapping the helper generator (DeepSeek-V3-Chat) to test dependence on its style; (iv) removing the refusal-template pairing (“I am sorry…”) to check if rote refusal is driving gains. Without these, attribution to skill coverage and generic data augmentation is uncertain.

3. The dictionary is built from single-turn pairs; yet, claims extend to multi-turn (GALA). While GALA's harmfulness decreases, the mechanism is not carefully analyzed (e.g., conversation-level skill composition, turn-ordering effects). A multi-turn extraction pipeline or at least qualitative mapping for sequences would improve the story.

**Questions:**

1. Table 3 labels the winning method SBAT, which appears to be ASCoT; if not a new baseline, this is a typo. Also, statistics (variance, CIs) and multiple-run stability for fine-tuning are not reported, which matters given sensitivity of safety tuning.

2. There are recent skill-mix efforts in instruction tuning and safety that could be directly compared as alternative composition engines rather than only CAT/LAT/WildJailbreak; the paper mainly positions against adversarial-training-style methods.

---

> ### Author Response · Authors · 2025-11-27
> **Response to Reviewer 5oss [Part-1]**
>
> We are grateful to the reviewer for their insightful feedback. The suggestions and comments have helped to improve its quality!
>
> > **Reviewer comment: The reviewer requests clearer quantification of the human-in-the-loop verification protocol used in skill extraction and scoring, expressing concern about potential bias or leakage from relying on GPT-4.1. They suggest adding independent human ratings or an additional non-OpenAI judge.**
>
> We thank the reviewer for raising this important concern regarding potential model-specific bias in the explainability evaluation. Following this feedback, we have strengthened the evaluation protocol in the revised draft. In addition to GPT-4.1, we now include a second independent judge- Claude 3.7 Sonnet- to rate the explainability of the sparse compositions using the same rubric and prompt templates. As shown in the updated Table 1, Claude’s scores closely mirror those of GPT-4.1 across all unseen attack families, and both judges agree on the relative ordering between Dfinal and Dover​. This cross-model alignment provides evidence that the observed explainability advantage of our compact dictionary is not an artifact of a single model’s reasoning style, and that our pipeline remains stable under distinct evaluators.
>
>
> > **Reviewer comment: The reviewer requests additional control experiments to isolate the source of ASCoT’s gains—specifically random-skill training, harmful-data-only training, swapping the helper generator, and removing refusal-template pairing—to ensure improvements are attributable to skill coverage rather than generic data augmentation.**
>
> (a) **Random Skill Control**
>
> We agree that comparing against a random skill subset is essential. In the revised draft, we introduce a new experiment (Appendix J) where we randomly sample 397 skills from the full pool of ~14k raw extracted skills- matching the size of our final dictionary- and generate training data using the same composition pipeline and budget. ASCoT outperforms this random-skill model:
>
> | Model                                 | PAIR ASR ↓ | AutoDAN-Turbo ASR ↓ |
> |---------------------------------------|-------------|-----------------------|
> | Base LLaMA‑3.1‑8B                     | 0.71        | 0.43                  |
> | Random 397 Skills                     | 0.18        | 0.14                  |
> | ASCoT (397 learned primitive skills)  | 0.11        | 0.07                  |
>
> The random subset underperforms because the raw pool is highly redundant: many entries correspond to near-duplicate micro-variants of the same manipulation (e.g., “euphemistic_keyword_translation’’ vs. “euphemistic_rewording’’). Random sampling therefore wastes capacity on repeated operations, producing low-diversity, low-coverage compositions.
> In contrast, our dictionary learning stage explicitly removes redundancy and identifies a sparse, distinct basis of adversarial primitives. This yields two benefits that random sampling cannot provide: (i) coverage- the ability to explain unseen attacks in terms of earlier ones; and (ii) targeted data generation- producing diverse, complementary compositions rather than repeating the same transformation. This principled compression is what enables ASCoT to achieve substantially better robustness than both the base model and the random-skill control.
>
> (b) **Training with the same data budget but more plain harmful data**
>
> In the revised draft, we have added a simple Refusal Training baseline that uses the same training data budget as ASCoT but replaces all composed skill-based examples with additional plain harmful prompts (without any compositional transformations). This baseline is now included in Table 2. The new Refusal Training baseline performs reasonably well on direct harmful requests but shows weak robustness to both seen and unseen jailbreaks, whereas ASCoT substantially outperforms it on all adversarial evaluations—confirming that simple data scaling with more plain harmful prompts does not replicate the gains from structured, skill-based compositional coverage.

---

> ### Author Response · Authors · 2025-11-27
> **Response to Reviewer 5oss [Part-2]**
>
> > **(Continued) Reviewer comment: The reviewer requests additional control experiments to isolate the source of ASCoT’s gains—specifically random-skill training, harmful-data-only training, swapping the helper generator, and removing refusal-template pairing—to ensure improvements are attributable to skill coverage rather than generic data augmentation.**
>
> (c) **Swapping the helper generator (DeepSeek-V3-Chat) to test dependence on its style**
>
> We selected DeepSeek-V3-Chat because it is strong enough to compose transformations, yet willing to produce harmful variants without refusal. These qualities make it particularly effective for generating diverse training prompts. That said, dependence on any single model’s stylistic tendencies is a valid concern.
>
> Importantly, our evaluation already tests cross-style generalization: the unseen attacks are produced by generators with very different stylistic profiles, including Mixtral-8×7B-Instruct (PAIR), GPT-4o-mini and GPT-3.5-turbo (PAP), Gemma-1.1-7B-IT (AutoDAN-Turbo), and GPT-4.1-mini (DarkCite, ImplicitRef). Despite this heterogeneity, ASCoT consistently outperforms baselines across all families. This shows that training on data composed using adversarial skill primitives generalizes across generator styles rather than overfitting to a particular composer’s style.
>
> (d) **Rote learning due to a fixed refusal template**
>
> To address this concern, we retrained all models using 20 stylistically diverse refusal templates (Appendix C). Despite removing any reliance on a fixed refusal phrase, ASCoT continues to outperform all baselines by a wide margin. This eliminates rote refusal as the source of its robustness and directly strengthens our central claim: ASCoT succeeds because its learned primitives provide broad compositional coverage of adversarial manipulations, not because it memorizes templated refusals.
>
> > **Reviewer comment: The reviewer requests to justify extending claims from single-turn skill extraction to multi-turn settings (e.g., GALA), noting that while harmfulness decreases, the underlying mechanism is not analyzed. They suggest adding a multi-turn extraction pipeline or qualitative mapping of skill sequences.**
>
> We thank the reviewer for this observation. We intentionally scoped our study to single-turn attacks because multi-turn jailbreaks involve mechanisms that extend beyond skill composition. In addition to adversarial skills, multi-turn attacks may succeed due to factors such as long-context attention dilution, conversational state manipulation, and cognitive-overload effects [r1, r2, r3]- dynamics that are not naturally representable as human-readable skill primitives and likely cannot be addressed through skill-level data coverage alone.
>
> That said, our inclusion of GALA results was a deliberate exploratory analysis to test whether single-turn skill coverage provides any transfer to multi-turn settings. The observed improvement (0.08 for ASCoT vs. 0.77 for the undefended model on Llama3.1-8B) suggests meaningful protection, even if other multi-turn-specific mechanisms remain unaddressed. A comprehensive treatment of multi-turn jailbreaks requires dedicated investigation, which we leave for future work. We will clarify this scoping decision in the revision.
>
> **References**
>
> [r1] Liu et al. Lost in the Middle: How Language Models Use Long Contexts. TACL 2024.
>
> [r2] Lu et al. LongSafety: Evaluating Long-Context Safety of Large Language Models. arXiv 2025.
>
> [r3] Anil et al. Many-Shot Jailbreaking. NeurIPS 2024.

---

> ### Author Response · Authors · 2025-11-27
> **Response to Reviewer 5oss [Part-3]**
>
> > **Reviewer comment: The reviewer requests us to report statistics such as variance, confidence intervals, or multi-run stability, given the sensitivity of safety fine-tuning.**
>
> To assess fine-tuning stability, we train ASCoT with three random seeds and report the mean and standard deviation of the Attack Success Rate (ASR) on PAIR and AutoDAN-Turbo. All seeds achieve very similar results and consistently outperform the base model, indicating that our conclusions are not sensitive to fine-tuning variability and remain stable across runs.
>
> | Model           | PAIR Strong Reject ↓ | AutoDAN-Turbo Strong Reject ↓ |
> |-----------------|------------------------|--------------------------------|
> | Base  (Llama 3.1 8b Instruct)          | 0.71                   | 0.43                           |
> | ASCoT (ours)    | 0.11 ± 0.02            | 0.07 ± 0.02                    |
>
>
> > **Reviewer comment: The reviewer suggests comparing ASCoT to more recent skill-mix or composition-based methods from instruction tuning and safety, rather than limiting comparisons primarily to CAT, LAT, and WildJailbreak.**
>
> We thank the reviewer for highlighting connections to recent compositional generalization studies in instruction following, robustness, and red-teaming [r1, r2, r3, r4, r5, r6, r7]. We see this comment as a valuable suggestion for positioning rather than a critique of novelty, and we appreciate the opportunity to clarify the relationship. Several conceptual differences distinguish our contribution:
>
> (1) **Bottom-up skill discovery vs. top-down skill mixing**: Many instruction-tuning works start from predefined or model-generated skills and then compose them top-down. In contrast, our pipeline is bottom-up: we extract skills directly from attack demonstrations and then learn a compact basis over them using sparse coding. This allows the dictionary to reflect empirical adversarial behavior rather than a manually curated or model-proposed set of skills.
>
> (2) **Coarse tactic bundling vs. fine-grained primitives**: Existing composition methods typically bundle coarse tactics (e.g., thematic transformations, high-level paraphrasing strategies). Our approach inserts an additional compression step that learns fine-grained, transferable adversarial primitives. This granularity provides both better interpretability and, as shown in our comparisons (e.g., with AutoDAN-Turbo), significantly stronger robustness.
>
> (3) **Static augmentation vs. temporal generalization**: Prior skill-mix methods focus on static augmentation—expanding training data with recombined variants. Our central goal is different: the Adversarial Déjà Vu hypothesis examines whether a principled skill basis learned from past attacks can explain and defend against future, unseen ones. This temporal framing drives our use of temporal cutoffs, dictionary learning, and sparse explanations.
>
> We have revised the Introduction section to explicitly cite these skill-mix efforts and clarify how our method builds on and extends this line of work.
>
> [r1] Yu et al. Skill-Mix: A Flexible and Expandable Family of Evaluations for AI Models. arXiv 2023.
>
> [r2] Park. Instruct-SkillMix: A Powerful Pipeline for LLM Instruction Tuning. Master’s thesis, Princeton University, 2025.
>
> [r3] Zhao et al. Can Models Learn Skill Composition from Examples? NeurIPS 2024.
>
> [r4] Jiang et al. WildTeaming at Scale: From In-the-Wild Jailbreaks to (Adversarially) Safer Language Models. NeurIPS 2024.
>
> [r5] Liu et al. AutoDAN-Turbo: A Lifelong Agent for Strategy Self-Exploration to Jailbreak LLMs. arXiv 2024.
>
> [r6] Doumbouya et al. H4RM3L: A Language for Composable Jailbreak Attack Synthesis. arXiv 2024.
>
> [r7] Wang et al. Lifelong Safety Alignment for Language Models. arXiv 2025.
>
> ***
> We appreciate your thoughtful feedback and hope our clarifications address the concerns raised. In addition to the substantive revisions, we have also fixed typos, improved formatting, and refined the overall structure and figures to strengthen the paper based on your suggestions. Should everything now be addressed, we would be grateful if you would consider adjusting your score accordingly.

---

### Official Review · Reviewer_DHsL · 2025-10-31

**Soundness:** 3
**Presentation:** 3
**Contribution:** 3
**Rating:** 8
**Confidence:** 4

**Summary:**

This paper proposes a new paradigm for the robustness of adversarial unknown jailbreak attacks, assuming that the new jailbreak attack is a combination of existing attack methods, and verifies it through investigating multiple papers. Based on this assumption, the paper proposes adversarial skill combination training called ASCoT, which improves model robustness through multiple adversarial attack combinations.

**Strengths:**

1. The hypothesis proposed in the paper is very novel, and it is very interesting and meaningful to verify it through proposed literature research.
2. The paper explains the skill-level overlap between seen and unseen jailbreak attacks by constructing interpretability scores and sparsity levels experiments.

**Weaknesses:**

1. Since seen attack methods are using automated processes in the research, it seems to rely on prompts, which may affect the stability of extracting skills.
2. The paper lacks extracting dictionaries from a subset of 32 research papers, forming a combination method, and experimentally verifying the gap in capabilities.

**Questions:**

Is there a space in the title of paper?

---

> ### Author Response · Authors · 2025-11-27
> **Response to Reviewer DHsL**
>
> We thank the reviewer for their thoughtful and constructive evaluation of our work. We are especially grateful for the recognition of the novelty of our Adversarial Déjà Vu hypothesis and the value of our literature-driven investigation across two years of jailbreak research.
>
> > **Reviewer comment: “Since seen attack methods are using automated processes in the research, it seems to rely on prompts, which may affect the stability of extracting skills.”**
>
> We thank the reviewer for raising this concern. Our seen-attack corpus contains both static jailbreaks and automated methods (e.g., multi-step prompt search, evolutionary prompt generators), and we agree that automated procedures can introduce run-to-run variability. To mitigate this, we do not rely on any single extraction trace; instead, we aggregate a large and diverse set of prompt–mutation pairs from each automated attack, which smooths out idiosyncratic noise.
>
> Most importantly, we assess stability through the behavior of the learned dictionary itself. As shown in Figure 3, when more seen attacks are incorporated, the explainability of unseen attacks steadily increases and then plateaus. This pattern can only occur if the dictionary is consistently capturing meaningful, invariant skill primitives rather than brittle artifacts of specific extraction runs. Early additions contribute genuinely new skills that raise explainability, whereas later additions have diminishing effect because the core stable primitive set has already been recovered. This rise-and-plateau trajectory provides empirical evidence of extraction stability.
>
> > **Reviewer comment: “The paper lacks extracting dictionaries from a subset of 32 research papers, forming a combination method, and experimentally verifying the gap in capabilities.”**
>
> We thank the reviewer for the comment. Because our goal is generalization to unseen attacks, our subset analyses must respect the temporal axis. Thus, instead of random subsets of the 32 papers, we construct temporal subsets.
>
> Our paper already includes this subset-based ablation in Figure 3: dictionaries learned from progressively larger temporal slices explain future unseen attacks increasingly well and eventually saturate.
>
> To provide more direct robustness evidence, we train two ASCoT-fine-tuned LLaMA-3.1-8B models using the same temporal cutoffs:
>
> | Temporal Cutoff | Num. Seen Attacks | PAIR ASR ↓ (Seen) | AutoDAN-Turbo ASR ↓ (Unseen) |
> |------------------|----------------|--------------------|-------------------------------|
> | Cutoff 1         | 7              | 0.115              | 0.225                         |
> | Cutoff 2         | 26             | 0.108              | 0.070                         |
>
>
> These results directly support the Adversarial Déjà Vu hypothesis: as we expand the set of seen attacks across the temporal axis, the apparent novelty of unseen attacks diminishes.
>
> ***
>
> We appreciate your thoughtful feedback and hope our clarifications address the concerns raised. In addition to the substantive revisions, we have also fixed typos, improved formatting, and refined the overall structure and figures to strengthen the paper based on your suggestions. Should everything now be addressed, we would be grateful if you would consider adjusting your score accordingly.

---

### Official Review · Reviewer_Q4KG · 2025-11-01

**Soundness:** 2
**Presentation:** 2
**Contribution:** 2
**Rating:** 2
**Confidence:** 4

**Summary:**

This paper proposes a new approach to improving the robustness of LLMs against jailbreak attacks. It introduces the Adversarial Déjà Vu hypothesis, which suggests that most new jailbreaks are not truly novel but rather combinations of previously observed adversarial “skills.” Authors analyze jailbreak studies over two years, extracting and compressing recurring manipulation strategies into a Jailbreak Dictionary. They claim that unseen attacks can often be explained as recombinations of these fundamental skills. Based on this insight, they propose Adversarial Skill Compositional Training (ASCoT), which trains models on diverse compositions of these skill primitives rather than isolated attacks.

**Strengths:**

1. The paper identifies a concrete limitation of existing adversarial training approaches—poor generalization to unseen jailbreaks—and motivates its method based on observed skill reuse patterns in prior attacks.
2. The paper is logically organized, with a smooth progression from hypothesis formulation to empirical analysis and method development, making it easy to follow and understand the core contributions.

**Weaknesses:**

1. The paper lacks concrete examples illustrating how the skill primitives from the Jailbreak Dictionary are applied to transform base prompts into adversarial ones. Since the proposed defense relies heavily on such compositions, the authors should include specific before-and-after prompt examples.

2. While the paper proposes a skill dictionary for generating jailbreak-style adversarial prompts to construct the ASCoT training data, the overall structure of using composed attack strategies for prompt generation resembles AutoDAN-Turbo, making the contribution feel incremental. To further clarify the contribution of the proposed Jailbreak Dictionary, authors should consider adding a comparative experiment against a baseline constructed using AutoDAN-Turbo strategies. Specifically, it would be informative to generate adversarial training data using AutoDAN-Turbo's discovered strategies and train a comparable defense model.

3. In the evaluation stage, AutoDAN-Turbo is claimed as one of the unseen attacks to test model generalization. However, according to Appendix C, the authors include the AutoDAN-Turbo paper within the curated set of 32 representative jailbreak papers used to construct their skill dictionary. Therefore, I do not think this can be called “unseen attack”. Authors should clarify the rationale and fairness of treating AutoDAN-Turbo as an unseen attack in the evaluation.

**Questions:**

See weakness.

---

> ### Author Response · Authors · 2025-11-27
> **Response to Reviewer Q4KG**
>
> Thank you for the thoughtful and constructive review. We appreciate the reviewer’s recognition that our work identifies a concrete limitation of current adversarial training approaches- namely, their poor generalization to unseen jailbreaks- and that our paper provides a clear and logically structured progression from hypothesis to analysis to method.
>
> > **Reviewer comment: The reviewer requests that we add concrete before-and-after examples showing how skill primitives are composed to transform a base prompt into an adversarial one.**
>
> We appreciate this suggestion. While we already provided qualitative examples in Appendix I.2 in the initial submission, we have now added an additional illustrative example in the main text (Figure 4) to show how skill primitives compose to transform a base prompt.
>
> > **Reviewer comment: The reviewer requests a clearer differentiation from AutoDAN-Turbo and suggests adding a comparative experiment where adversarial training data is generated using AutoDAN-Turbo’s strategy set, enabling a direct baseline comparison.**
>
> We thank the reviewer for this thoughtful suggestion. We respectfully disagree that our contribution is incremental relative to AutoDAN-Turbo, as the two works differ fundamentally in objectives and methodology.
>
> **(1) Different objectives**: AutoDAN-Turbo is an attack algorithm that adaptively reuses tactic groups successful against specific target models, aiming to maximize attack success rate. Our work addresses the complementary defense problem: how to proactively defend against attacks that do not yet exist. The Adversarial Déjà Vu hypothesis—that future jailbreaks are largely recombinations of skill primitives from past attacks—provides a principled foundation for this goal. Our temporal cutoff study (Section 2.4, Figure 3), which demonstrates monotonically increasing explainability as skill coverage grows, is the first empirical validation of this hypothesis across a two-year attack corpus.
>
> **(2) Different methodology**: Our pipeline introduces several novel technical components absent from AutoDAN-Turbo, which are essential to validate the hypothesis:
>
> - (a) Sparse dictionary learning compresses ~14,000 raw extracted skills into ~400 transferable primitives, enabling systematic coverage of the adversarial skill space.
>
> - (b) LLM-augmented basis pursuit (Eq. 2–3) provides interpretable decomposition of new attacks into weighted combinations of known primitives, enabling both explainability analysis and principled training data generation.
>
> In summary, skill extraction and composition are just one part of our proposed pipeline for validating the hypothesis. Even zooming into the composition module alone, we differ substantially from AutoDAN-Turbo: it uses coarse tactic bundles discovered during attack optimization (e.g., "Defensive Education Justification & Fictional Narrative Abstraction"), whereas we learn fine-grained, transferable primitives (e.g., "academic_research_pretexting," "persona_override_roleplay") from cross-attack patterns via dictionary learning. This finer granularity enables better explainability and systematic coverage through composition.
>
>
> **Experimental validation**: To address the reviewer's suggestion directly, we trained a defense model using AutoDAN-Turbo's tactic groups (matched in data size). On the PAIR StrongReject benchmark:
>
> | Model                        | Harmfulness Score ↓ |
> |------------------------------|----------------------|
> | AutoDAN-Turbo-trained model | 0.20                 |
> | ASCoT-trained model         | 0.11                 |
>
>
> Training on our compact skill primitives provides substantially stronger robustness, underscoring the effectiveness of our proposed pipeline.
>
> > **Reviewer comment: The reviewer asks us to clarify why AutoDAN-Turbo is treated as an unseen attack during evaluation**
>
> Thank you for the opportunity to clarify this point. Appendix C lists all 32 jailbreak papers surveyed, not the subset actually used to build the Jailbreak Dictionary. As explained in Sections 2.4 and 3.2, we apply a strict temporal cutoff of August 15, 2024: attacks before this date are treated as seen; those after are unseen. AutoDAN-Turbo was released in October 2024, and therefore was not included in dictionary construction and is correctly evaluated as an unseen attack.
>
> ***
>
> We appreciate your thoughtful feedback and hope our clarifications address the concerns raised. In addition to the substantive revisions, we have also fixed typos, improved formatting, and refined the overall structure and figures to strengthen the paper based on your suggestions. Should everything now be addressed, we would be grateful if you would consider adjusting your score accordingly.

---

> > ### Comment · Reviewer_Q4KG · 2025-11-27
> >
> > I thank the authors for their comprehensive and thoughtful response.
> >
> > 1. The addition of Figure 4 in the main text significantly improves clarity. The before-and-after prompt transformations now make the skill composition mechanism tangible and understandable.
> >
> > 2. I am now convinced the contribution is not incremental. The authors' response effectively clarified three key distinctions: objective difference, technical novelty, and empirical validation.
> >
> > 3. The clarification about the August 15, 2024 cutoff fully resolves my concern. AutoDAN-Turbo (October 2024) was correctly treated as unseen.
> >
> >  Remaining Minor Suggestions:
> >  - The comparative experiment with AutoDAN-Turbo could be highlighted more prominently

---

### Official Review · Reviewer_jXni · 2025-11-02

**Soundness:** 2
**Presentation:** 3
**Contribution:** 3
**Rating:** 6
**Confidence:** 4

**Summary:**

This paper proposes an efficient and effective adversarial training (AT) approach for LLMs. The main hypothesis of the approach is that each new jailbreak attack can be implicitly seen as a composition of previously existing attacks. Under this hypothesis, the authors design a novel method to compress existing jailbreak attacks into a small set of "seed attacks", and then use these seed attacks to efficiently generate new strong jailbreak attacks for AT. These compressed seed attacks are also used to analyze the evolution of real-world jailbreak attacks.

**Strengths:**

1. The core hypothesis of this paper is that existing jailbreak attacks can be implicitly seen as compositions of previous jailbreak attacks. The authors also design a pipeline for decomposing given attacks. I like this idea and believe it is novel and may potentially help the LLM safety community understand more about the mechanisms behind LLM jailbreaking.

2. The authors design a smart pipeline to reconstruct human-readable attack skill descriptions from the compressed jailbreak dictionary embeddings $D$ obtained from Eq. (1) (which is achieved via Eq. (2) and the help of LLM prompting). I really like this solution.

**Weaknesses:**

1. **Too few base models.** The AT experiments in Section 3 were only conducted on two LLMs (Llama3.1-8B, Zephyr-7B), which I think is far from enough to justify the effectiveness of the proposed Jailbreak Dictionary method in mitigating attacks. Experiments on more model families such as Qwen2.5/3, Gemma-2/3, Llama-2, Vicuna, and Mistral are suggested.

2. **Missing important representative jailbreak baselines.** The authors evaluate their AT approach against four unseen attacks, but most of these attacks (except AutoDAN-Turbo) are not representative attacks that are widely adopted as jailbreak baselines for jailbreak robustness analysis. Therefore, I suggest the authors adopt more representative attacks such as [r1, r2, r3] as unseen attacks.

3. **Missing important details about jailbreak robustness evaluations.** For Table 2 in Section 3.2, please clarify:
    - What is "StrongReject score"? Is it a variant of the "Attack Success Rate"?
    - On which dataset is the "StrongReject score" calculated?
    - How is the "StrongReject score" calculated?

4. The "prompting-based skill extraction" technique in Section 2.2 and the "attack via prompting with skill descriptions" technique in Section 3.2 are not new. To the best of my knowledge, these methods were first proposed in the AutoDAN-Turbo paper in 2024. I think the authors should explicitly acknowledge this in the main text of their paper.

5. The format of this paper is incorrect. The width of this paper is far larger than the restriction of the ICLR submission template. The authors must fix this in their revision.



**References**

[r1] Hayase et al. Query-Based Adversarial Prompt Generation. NeurIPS 2024.

[r2] Sadasivan et al. Fast Adversarial Attacks on Language Models In One GPU Minute. ICML 2024.

[r3] Andriushchenko et al. Jailbreaking Leading Safety-Aligned LLMs with Simple Adaptive Attacks. ICLR 2025.

**Questions:**

See **Weaknesses**.

---

> ### Author Response · Authors · 2025-11-27
> **Response to Reviewer  jXni [Part-1]**
>
> We are grateful to the reviewer for their insightful feedback. The suggestions and comments have helped to improve its quality!
>
> > **Reviewer comment: The reviewer requests that we include ASCoT experiments on a broader set of base models beyond Llama-3.1-8B and Zephyr-7B.**
>
> We thank the reviewer for this valuable suggestion. In response, we have expanded our study to include an additional, widely used model family: Mistral-7B-Instruct-v0.2. The updated Table 2 now reports full results for this third model family. Across all evaluation attacks- including unseen ones-our ASCoT method continues to outperform all baselines. These expanded results further strengthen the evidence that our approach generalizes well across diverse model architectures.
>
> > **Reviewer comment: The reviewer requests that we evaluate ASCoT against additional and more widely adopted jailbreak baselines beyond the four unseen attacks currently included.**
>
> We thank the reviewer for the helpful suggestions. As clarified in the introduction—specifically at line 70 of the paper—our work focuses on unseen, language-based black-box attacks, where adversaries interact with the model solely through natural language:
>
> *“In contrast, attacks that exploit internal access to the model—such as fine-tuning (Qi et al., 2023)—may not decompose into reusable language skills.”*
>
> The reviewer-suggested attacks [r1–r3] require internal access in terms of log-probs and therefore fall outside the setting our compositional hypothesis is designed to model. Nonetheless, for completeness, we have incorporated BEAST [r2] and the Simple Adaptive Attack [r3] into our evaluation (an open-source implementation for [r1] was not available). The updated Table 2 now reports results on these attacks. Importantly, ASCoT remains robust even against these non-language-based attacks, despite being trained exclusively on natural-language skill compositions, further underscoring the strength of our approach.
>
> **References**
>
> [r1] Hayase et al. Query-Based Adversarial Prompt Generation. NeurIPS 2024.
>
> [r2] Sadasivan et al. Fast Adversarial Attacks on Language Models In One GPU Minute. ICML 2024.
>
> [r3] Andriushchenko et al. Jailbreaking Leading Safety-Aligned LLMs with Simple Adaptive Attacks. ICLR 2025.

---

> ### Author Response · Authors · 2025-11-27
> **Response to Reviewer jXni [Part-2]**
>
> > **Reviewer comment: The reviewer asks for clarification about the definition, dataset, and computation procedure of the “StrongReject score” reported in Table 2.**
>
> We would like to clarify the evaluation procedure used for StrongReject. StrongReject is a variant of Attack Success Rate introduced in [r4], which jointly measures a model’s willingness and capability to produce harmful content. As defined in the benchmark, each model response is scored using a rubric that combines:
>
>  (1) a binary refusal indicator,
>
>  (2) a specificity score, and
>
>  (3) a convincingness score,
>
> yielding a continuous harmfulness value in [0,1].
>
> All StrongReject results reported in Table 2 are computed using the official StrongReject forbidden-prompt dataset. For each jailbreak method, we apply the transformation to the harmful instructions and score the resulting outputs using the rubric-based evaluator, with OpenAI GPT-4.1-mini as the judge- exactly as specified in the benchmark. This ensures consistency and fair comparison across all attacks and model families. We have updated the paper to explicitly clarify these details.
>
> **References**
>
>  [r4] Souly et al., A StrongReject for Empty Jailbreaks, NeurIPS 2024.
>
>
> > **Reviewer comment: The reviewer requests that we explicitly acknowledge prior work-specifically AutoDAN-Turbo-as earlier introducing prompting-based skill extraction and prompting attacks using skill descriptions.**
>
> We thank the reviewer for highlighting this point. We fully agree that prompting-based skill extraction and recombination were explored earlier, most notably in AutoDAN-Turbo [r5] and WildTeaming [r6], and we now explicitly acknowledge this in the Introduction.
>
> Our contribution, however, is not the prompting mechanism itself but the full adversarial skill pipeline and the Adversarial Déjà Vu hypothesis. Prompting is only one step in a broader framework: we extract skills from 32 papers over two years, apply dictionary learning to compress thousands of noisy skills into ~400 transferable primitives, and use these primitives to study temporal coverage, quantify novelty via basis pursuit, and train proactive defenses.
>
> Prior prompting-based works do not learn a reusable skill basis or analyze temporal generalization or explainability. Thus, while our extraction step builds on earlier ideas, the overall hypothesis, methodology, and objectives represent a distinct and novel direction.
>
> **References**
>
> [r5] Liu et al. AutoDAN-Turbo: A Lifelong Agent for Strategy Self-Exploration to Jailbreak LLMs. 2024.
>
> [r6] Jiang et al. WildTeaming at Scale: From In-the-Wild Jailbreaks to (Adversarially) Safer Language Models. NeurIPS 2024.
>
>
> ***
>
> We appreciate your thoughtful feedback and hope our clarifications address the concerns raised. In addition to the substantive revisions, we have also fixed typos, improved formatting, and refined the overall structure and figures to strengthen the paper based on your suggestions. Should everything now be addressed, we would be grateful if you would consider adjusting your score accordingly.

---

### Author Response · Authors · 2025-12-02
**Discussion-Period Summary for ACs, SACs and PCs [Part-2]**

## Addressed Reviewer Concerns

**(A) Expanded model families:** Added Mistral-7B-Instruct-v0.2 at the request of Reviewer jXni to test architecture-level generality. ASCoT continues to outperform baselines across all attack suites.

**(B) Expanded jailbreak baselines:** Added BEAST and the Simple Adaptive Attack. Reviewer jXni suggested evaluating more representative attacks. ASCoT remains robust even to these stronger, non-language-based attacks.

**(C) Key control experiments added (Reviewer 5oss):** At the suggestion of Reviewer 5oss, we added several key control experiments to isolate the source of ASCoT’s gains. First, we trained a model using a random-skill subset matched to the dictionary size, which performed substantially worse- demonstrating that principled dictionary learning, not raw scale, drives robustness. Second, we added a harmful-data-only (Refusal Training) baseline with the same training budget but without any compositional transformations; its weaker performance confirms that adversarial skill coverage, rather than simply adding more harmful prompts, is essential. Finally, we introduced 20 diverse refusal templates to ensure that robustness is not explained by memorization of a single refusal pattern.

**(D) Fine-tuning stability:** Three-run seed stability added per Reviewer 5oss. ASCoT shows low variance across seeds on both PAIR and AutoDAN-Turbo.

**(E) Explainability robustness across evaluators:** Added Claude 3.7 Sonnet as a second independent judge. Reviewer 5oss raised concerns about GPT-4.1 dependence; Claude closely mirrors GPT-4.1’s rankings, confirming evaluator-agnostic explainability trends.

**(F) Concrete examples of skill composition:** Reviewer Q4KG requested explicit before/after transformations. We added illustrative examples of compositional attack generation in the main text (Figure 4).

**(G) Clarification of evaluation details:**  We also expanded several evaluation clarifications requested by the reviewers. We now provide a clear description of the StrongReject rubric, the dataset on which it is computed, and the associated scoring protocol. We further clarified the temporal cutoff (August 15, 2024) that separates seen from unseen attacks in our analysis.

**(H) Distinction from Prior Skill-Mixing Methods:** We also clarified how our pipeline fundamentally differs from AutoDAN-Turbo and prior skill-mixing approaches. Our method builds a defense aimed specifically at unseen future jailbreaks. Methodologically, we introduce dictionary learning, LLM-augmented basis pursuit, and temporal analysis, none of which appear in prior pipelines. Finally, whereas earlier methods rely on coarse tactic bundles, our bottom-up extraction yields fine-grained adversarial primitives that better span and explain the underlying skill space.

---

**We hope the above summary is helpful in your assessment and shows that reviewers’ concerns have been addressed while preserving the main contributions of our work. We appreciate your consideration.**

Warm regards,

The authors of submission 21925

---

### Author Response · Authors · 2025-12-02
**Discussion-Period Summary for ACs, SACs and PCs [Part-1]**

Dear ACs, SACs, and PCs,

Thank you for overseeing our submission. We appreciate the reviewers’ thoughtful feedback, which has helped us further strengthen and clarify our work.

## Reviewer Ratings & Discussion Outcomes

Before the rebuttal, **three reviewers provided positive evaluations (scores 6, 6, and 8)**. The remaining reviewer (Q4KG) initially gave a 2, but after our clarifications **explicitly stated that their concerns were fully resolved and raised their score to 6**. Due to the shortened discussion window, we were unable to hear further from the other reviewers; however, all their concerns were requests for clarification, extended evaluation, or minor fixes, which we addressed comprehensively. **No core changes were made to the method or claims, and the main contributions remain intact.**

## Core Novel Contributions

**(1) Adversarial Déjà Vu Hypothesis:** Our work introduces and validates a novel hypothesis- future jailbreaks are largely sparse recombinations of adversarial skills extracted from past attacks. **Reviewer jXni and Reviewer DHsL** both highlighted this as a meaningful and novel direction. We conduct the first large-scale temporal analysis of jailbreaks- covering 32 attack papers spanning two years- and show that as more past attacks are incorporated, the ability to sparsely explain unseen attacks consistently rises and eventually saturates. This reveals that the apparent novelty of future jailbreaks diminishes over time.

**(2) Jailbreak Dictionary Learning Pipeline:** Reviewers commended our bottom-up extraction of thousands of skills across 32 jailbreak papers and the compression of these into a compact dictionary of adversarial primitives. This dictionary provides a principled basis spanning the adversarial skill space, rather than coarse tactic bundles. **Reviewer 5oss** highlighted our explainability experiments and the strength of the compact dictionary compared to the overcomplete version.

**(3) LLM-Augmented Basis Pursuit for Explainability:** **Reviewers (jXni and 5oss)** specifically praised our use of the LLM-assisted Basis Pursuit algorithm to generate interpretable explanations of unseen attacks- calling it a “smart” and “novel” technique. This mechanism provides a new lens for understanding jailbreak evolution through the learnt Jailbreak Dictionary.

**(4) Adversarial Skill Compositional Training (ASCoT):** **Reviewer Q4KG** noted that generalization to unseen attacks is a critical open problem in safety. Our method directly targets this by training on skill-level compositions rather than isolated attacks. Across all model families and attack suites- including new baselines added after reviewer suggestions- ASCoT consistently improves robustness without harming utility. We further study robustness through the lens of adversarial skills and uncover key phenomena such as the Coverage Dividend and an optimal compositional depth that maximizes safety gains.

---

### Meta-Review · Area_Chair_1jTh · 2025-12-31

**Summary:**

The paper's proposed methodology is interesting. I am personally very partial to the idea of using DL for this task and find it principally justified. That said, I think the paper should better show the advantages of using DL over other methods, such as clustering or simple deduplication. That said, it is clear that methodologically showing how to combine DL and language/LLMs is interesting and has applications beyond jailbreaks. Further, the DejaVu observation in itself is an important contribution of the paper that I believe will have wide implications throughout the community. I also think it will be a nice measurement of the novelty of new jailbreak attacks, as I think it is widely recognized in the current field that many jailbreak methods (especially the very narrow ones) have become hard to keep track of and are of limited scientific value. All in all, the paper's merits outweigh by a lot the missing opportunities to convince the reader of the effectiveness of the methodology.

**Reviewer Concerns:**

**Outstanding reviewers' concerns**
- **Reviewer jXni (Weakness 2)**
One outstanding question here is that the authors add BEAST and the adaptive attack in the seen category, but I think the reviewer wanted them in the unseen one. I think, more generally, it is interesting to see which of the 32 attacks truly advanced the field (in the sense of Figure 3) and whether a small subset of "representative" papers for comparison can be extracted, that can be adapted as a "standard set" of attacks people should use to compare their defenses. The authors is encouraged to try to address some of these questions in their final manuscript.
- **Reviewer jXni (Weakness 5)**
While I didn't have the opportunity to verify this (I am only allowed to see the revised version of the paper), this is potentially a very serious breach of the rules of ICLR. I employ the authors to ensure that the final version of the paper follows the paper template very closely. Further, I employ authors to do that for all their future submissions, too. **You can legitimately be desk rejected for this**, and nothing will pain me more than having to desk reject a legitimately good paper such as this one.
- **Reviewer 5oss (Weakness 2)**
I think one aspect of this that the authors do not address is the composition of their chosen training set described in **Training Dataset Construction** in the paper. I would love to see some small-scale experiments justifying the need for each of the components of the dataset.

**AC's questions and concerns**
- **$D_\text{over}$ is not properly defined anywhere in the paper**
Please include proper definition for $D_\text{over}$ in Table 1.
-  **CAT\* and LAT\* 's 40K samples are not detailed**
Please clarify in the final revision if the 40K samples are produced with the CAT and LAT's original algorithms
- **What kind of skills were extracted from GCG?**
The prompts used seem good for describing interpretable attacks like the ones in classical jailbreaks. I wonder what skills were extracted by less interpretable attacks like GCG. An appendix with examples would be nice.
- **The authors report avg ASR across attacks in Table 2, but ALO is also interesting**
[1] Introduces ALO (at least one) ASR measurement as a better way to combine the ASR of different attack strategies. I think adapting it (in addition to the avg ASR) will be nice in reporting the results in Table 2.
- **Is dictionary learning truly needed?**
The authors mention that clustering with embedding models is an alternative to their dictionary learning scheme to reduce the redundancy of their initial skill set. I think a further comparison and discussion in the paper will strengthen the evidence that the technical complexity of the dictionary recovery is needed. In particular, finding the size of the dataset reduced through clustering/embedding distances that match the DL approach and showing that it is substantially larger will be very useful. I also find [2] to be somewhat more generic but related to this work, as this work essentially is about efficiently creating a taxonomy of jailbreaks.
- **MixAT[1] makes for a nice baseline**
I think MixAT[1] could be introduced as a baseline, on top of CAT and LAT, as it focuses specifically on generalizability of adversarially trained models.
- **Please incorporate all the additional experiments and discussions from the rebuttal in the final paper**
Quite a lot of new experiments and clarifications were provided in the rebuttal. I strongly encourage the authors to incorporate them in the final manuscript.

[1] https://arxiv.org/abs/2505.16947
[2] https://arxiv.org/pdf/2403.12173

**Reviewer Scores:**

- **Reviewer jXni**
If I were the reviewer, I would have raised the score to an 8.
- **Reviewer Q4KG**
I think the reviewers' own proposed 6 is reasonable.
- **Reviewer DHsL**
The review was not considered due to its poor quality. Yet, I do think some of the authors' rebuttal comments are surprisingly useful.
- **Reviewer 5oss**
If I were the reviewer, I would have raised the score to an 8.

---

### Decision · Program_Chairs · 2026-01-26

Accept (Poster)